# The Reflection of Income Segregation and Accessibility Cleavages in Sydney's House Prices

**Matthew Kok Ming Ng *** , **Josephine Roper** , **Chyi Lin Lee and Christopher Pettit**

City Futures Research Centre, University of New South Wales, Sydney, NSW 2052, Australia;
j.roper@unsw.edu.au (J.R.); chyilin.lee@unsw.edu.au (C.L.L.); c.pettit@unsw.edu.au (C.P.)
* Correspondence: matthew.ng@unsw.edu.au

**Abstract:** Cities often show residential income segregation, and the price of housing is generally related to employment accessibility, but how do these factors intersect? We analyse Greater Sydney, Australia, a metropolitan area of 5 million people. Sydney is found to have reasonably even employment accessibility by car, reflecting the increasingly polycentric nature of the modern city; however, it also shows considerable income segregation and variance in property prices between different parts of the city. Entropy is used to examine diversity and mixing of different income groups. Finally, hedonic price models using ordinary-least squares and geographically-weighted regression techniques show the differing effects of employment accessibility on house prices in different parts of the city. The results show that accessibility has small to negative effects on prices in the most valuable areas, suggesting that other effects such as recreational access and employment type/quality may be more important determinants of house prices in these areas.

**Keywords:** accessibility; geographically-weighted regression; house price; gravity models; hedonic price models

## 1. Introduction

Disparities in access in cities imply a spatial mismatch of services, activities, and populations that may compound and enforce inequality [1–4]. Unequal accessibility to employment particularly has tangible manifestations within a city's socioeconomic structures [1]. This has disproportionate effects on low-income and ethnic groups [1], on the availability of affordable housing [5,6], the reinforcement of poverty cycles [4,7], and the differential outcomes of demographic groups that persist transgenerationally [8].

Understanding the degree to which variations in accessibility are seen within cities, and the way these variations are spatially distributed, forms the basis of this paper's inquiries. In particular, it questions how accessibility may relate to variations in local income distribution and house prices. Property prices enumerate both individual property characteristics and locational attributes into a single figure. A key part of these locational attributes is the area's accessibility [9,10]. With this, both incomes and property prices can be conceptualised as a component of a city's organisation. This paper addresses the question of whether accessibility is considered in the valuation of more highly-priced areas, and, if so, whether high-accessibility areas remain exclusively for a wealthier demographic group.

Our city of focus, Sydney, Australia, is considered one of the least affordable cities in the world [11], and segregation is increasing in Sydney [12]. As such, it is important to have an enhanced understanding of the nexus between accessibility, income segregation, and house prices in this area given the paucity of existing research. Further, in the framework of the United Nations' goal of achieving more inclusive economic development in cities, Sustainable Development Goal 11, understanding these dynamics may contribute more widely to more targeted housing policy and planning measures, as well as interventions.

## 2. Research Objectives

The next section presents the previous relevant work and the theoretical framework (Figure 1) for this paper's analysis of accessibility and income segregation. Then, the Methods addresses three research aims. First, the spatial distribution of income groups in Sydney is quantified and visualised to highlight whether instances of income segregation are apparent. Here, entropy statistics are used as a measure of income diversity within areas; and their relationships to the relative proportion of different income groups are considered. Next, these income groups are analysed with respect to the spatial variations to employment accessibility. As discussed above, poor accessibility is a barrier to upward mobility; therefore, determining the effects on lower-income groups is fundamental to achieving urban and social equity. Lastly, the above three variables are tested with respect to house prices in Greater Sydney to uncover where urban disparities manifest themselves with respect to residential property values.

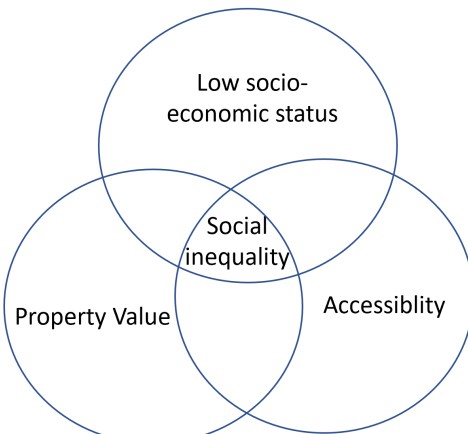

**Figure 1.** Proposed factors contributing to social inequality in Sydney.

## 3. Literature Review

### 3.1. Accessibility

Accessibility is a vital yet complicated component of the built environment. It is a measure that arises from combining the dimensions of land use, network structures, and population distribution [13] to describe how the population can reach the land uses using the network. It is used to explain and describe urban mobility within city systems, which include aspects of locational planning and facility distribution, equitable urban growth, as well as being used as a quantifiable metric for policy decisions [14–17]. Its common inclusion in these studies reflects its power in representing complex urban processes into an interpretable metric across different geographies [18,19]. This is evidenced from the manifold studies that have now operationalised accessibility analyses to discuss issues of resource and infrastructure provision [20], and land-use decision-making [21–24]; systemic concerns of urban equity [20,25], social segregation [26–28], and planning for alternative development futures [29,30]. Given its importance in cities, understanding accessibility thus provides a powerful means to assess and interrogate policies of urban growth.

Many studies have demonstrated how land and property prices adjust to account for the changing spatial distributions of urban facilities and activities [31,32]. More recently, in Australia, Pettit et al. [10] and Lieske et al. [33] have spearheaded research endeavours in this domain. Both have highlighted the profound impact of policy and city infrastructure decisions on property values in the real world; and their work in enumerating the contributions of individual characteristics and neighbourhood variables to property values has had many implications on the use of quantifiable data to substantiate policy-making decisions. Similarly, earlier work by Mulley [9] investigated the link between accessibility and transport modalities in creating property value changes in Sydney. It was

found that proportionality in these variations was not uniform and was changeable across different locations.

In the same vein, this paper aims to expand the discourse through the addition of accessibility as a variable that is complementary to those used by Pettit et al. [10] and Mulley [9]. Employment accessibility is considered specifically, given its demonstrated links to economic productivity and upward mobility [1,34–37]. An important contribution for the analysis conducted in this paper is the consideration of social inequality by examining Sydney's neighbourhood income stratum and how this is reflected in property values.

*3.2. Income Segregation and House Prices*

The above section has briefly discussed the many effects of accessibility. Thematically, the discussion presented issues of persistent segregation, land-use constraints, and hurdles for upward mobility amongst disadvantaged demographic groups. These are a few themes noted most prominently in the existing literature [38,39]. Indeed, differential accessibility and its varied impacts on city form and function is important to understand for numerous reasons. Geographies are not homogeneous, and existing hyperlocal variations have far-reaching implications on both the structure of the urban system and the livelihood of its residents [25]. Banister [40] and Farrington [41] noted that deft urban governance requires a granular appreciation of these hyperlocal variations to articulate the swathe of socioeconomic imbalances, arguably brought on by decreased accessibility. They raise the idea of a 'poverty of access' as a tangible urban condition. Addressing this is a key challenge for all levels of governance and city planning [41] (p. 324).

Central to this is the notion that poor accessibility comes in tandem with increased isolation. Variations in the spatial distribution of resources, essential amenities, and critical urban functions cause disparities that may work to simultaneously offer or restrict, opportunities (i.e., employment, education, healthcare, recreation) to specific population groups simply by virtue of their location [25,42–44]. Consequently, differential costs are incurred across the urban system. Groups living in low-accessibility areas are often disproportionately hampered through a multitude of costs, which include the increased financial expenditure of extended travel, in addition to critical losses in time—and opportunity—costs to overcome these shortfalls [34,45–48].

Differential costs are most apparent when low-accessibility is also coupled with lowered income [49]. Taking the specific example of access to employment opportunities, it was found that the differential costs of movement incurred by segregated, low-income communities have the adverse effect of restricting the search for employment to proximate areas, despite the availability of higher potential incomes further away [49]. Separating distances play a large role in these choices, particularly as these demographic groups often require greater ease to travel given the ad hoc nature of their employment [49–51]. Giuliano and Small [51] also reported that employing institutions often do not regard commuting distance as a barrier to workforce hires. Hence, there is typically no strong impetus to increase job proximity through the spatial redistribution of employment. Coupled with lowered incomes, it becomes clear that the issue of poor accessibility can be all the more punishing. Its prolonged effects in urban pockets often precedes the ghettoisation of lower-income earners [34,52].

The use of property values may be useful in examining these issues in finer detail. House prices plays an important role in contributing to the isolation of demographic groups [53]—in that, where affordable housing is in a deficit, segregation is exacerbated and access into better serviced neighbourhoods becomes unattainable or difficult. Moreover, the increasing concentration of wealth that is accumulated from home ownership brings with it significant intergenerational benefits [54]. This can then further reduce affordability in these urban pockets [52], and with the disproportionate rise in wealth over time, the same issues of inequality are repeated. It is worth exploring whether house prices can be used as a proxy to determine both income and accessibility disparities within neighbourhoods.

Indeed, few studies have investigated these specific relationships, despite being a long-standing item of debate [55].

### 3.3. Income and Wealth Inequalities in Australia

In [56], the Australian Government's Productivity Commission released a report that examined inequality—with the distribution of income and wealth being most prominent—over a period between 1988 and 2016. The report concluded that inequality in Australia had only a marginal increase over the past three decades, as alleviated by Australia's sustained economic growth [56]. It was further posited that significant growth was seen across all analysed socioeconomic groups; and with disparities seen at a reduced level in comparison to other developed nations [56]. Since its release, however, reports have challenged these findings [55,57,58]. They caution an oversimplification and a downplay of these disparities, which obscures the real complexity in the challenges of inequality faced by Australians.

In a more detailed representation of this inequality, the increase in disposable incomes within the country's highest earners has now reached a factor of over 25 in comparison to the lowest 5 per cent income bracket [55]. Income disparities were also noted to be more unequal than global metrics published by the Organisation for Economic Cooperation and Development (OECD) [55]. These criticisms come on top of those put forth by Gittins [59], who established that poverty rates in Australia actually exhibited negligible improvements despite the growth in real incomes. A total of 40 per cent of Australia's lowest income earners depend on social welfare, in addition to low—full-and part-time—employment [55]; whereas, in contrast, the wealth in Australia's upper echelons of income earners more than quadrupled [42].

It was proposed that a large part of these discrepancies was due to large increases in house prices driven by capital investments in the mid-2000s [55,58]. This magnified wealth inequality in the country as upper—and middle—class groups, who have more capital to invest in property, saw increases in wealth, while lower-income households had negligible gains in wealth [59]. Australia's highest socioeconomic groups hold, on average, approximately AUD 1.95 million in assets, in comparison to the AUD 0.3 million in lower income groups [55]. For higher income earners, 80 per cent of their wealth is comprised of properties as assets, with approximately 40 per cent attributed as a main residence, and an additional 12 per cent classed as investment real estate [42,58]. These figures come in stark contrast to the 16 per cent average held by lower income groups on total property [55].

It is clear from the findings above that the major role that property plays in the distribution of wealth in Australia requires further investigation. Perhaps what is most striking in the research undertaken by Davidson et al. [55] is the extent to which lower-income households are dependent on part- and full-time employment, as well as welfare payments, in comparison to more passive income streams available to those upper socioeconomic groups through assets. For urban governance to facilitate changes in policies to redress these inequalities, governing bodies must understand local variations in access to jobs and area-level employment, in addition to how these factors spur socioeconomic segregation given their disproportionate importance to lower-income groups. This further allows considering more systemically how the built environment can be best shaped from its current state to uplift these communities. Certainly, these issues are all multifaceted and perhaps cannot be fully represented within any one urban metric; however, access to employment forms a basis from which policies can be tested, interrogated, and revised to offer the best outcomes for all population groups.

## 4. Data

A number of datasets were used to examine above relationships. Table 1 provides a list of these datasets and their sources. These are expanded upon in the relevant subsections with respect to their specific use as components in downstream analyses.

**Table 1.** Table of datasets utilised and their sources.

| Dataset | Source |
|---|---|
| Road Network | OpenStreetMap |
| Statistical Area 2 Neighbourhood Attributes | Australian Bureau of Statistics |
| Journey to Work | Australian Bureau of Statistics |
| Income Data | Australian Bureau of Statistics |
| Geographic Statistical Boundaries | Australian Bureau of Statistics |
| House Price | Australian Property Monitor |
| Points of Interest (including building footprints) | Geoscape Australia |

### 4.1. Census Data

Census datasets from the Australian Bureau of Statistics (ABS), which include neighbourhood variables on median incomes and educational attainment, as well as employment and crime rates, were used. These neighbourhood indicators are available at both the Statistical Area 1 (SA1) and Statistical Area 2 (SA2) level from the ABS. SA1s, the primary level of analysis in this paper, are variable in size with a median population of 420 people, and area of 0.125 km$^2$ in Greater Sydney. In addition to the above datasets, more granular data on personal income, and commuting flows were also obtained from the ABS. These form integral components to the accessibility modelling, and their use is discussed in the following subsections.

#### 4.1.1. Personal Income Data

Income data for the 2016 census year was obtained at the SA1 level for the entirety of the Greater Sydney metropolitan area. The dataset refers to the ABS' Total Personal Income dataset, INCP, which collects the gross personal income received each week by an individual [60]. The data was filtered by an individual's place of usual residence to to reflect their residential locations. The INCP provided 17 income groups that ranged between AUD 0 to over AUD 3000 per week. These ranges were aggregated into three relative income groups, with income threshold adapted from the Australian Council of Social Services [61] and the Property Council of Australia [62]. Low-income groups in this study are defined as individuals with a personal income of less than AUD 649 per week; middle-income groups ranged between AUD 650 to AUD 1999 per week; and high-income groups were those with incomes above these thresholds. The distributions of these income ranges are illustrated in Figure 2. Both the total income groups and and their relative census tract proportions are taken into consideration. A total of 1.5 million low-income—($\bar{x}_{l\_tract} = 0.471$), 1.3 million middle-income—($\bar{x}_{m\_tract} = 0.376$), and 0.4 million high-income—($\bar{x}_{h\_tract} = 0.153$) individuals were enumerated in the dataset bringing the dataset size to approximately 3.2 million individuals.

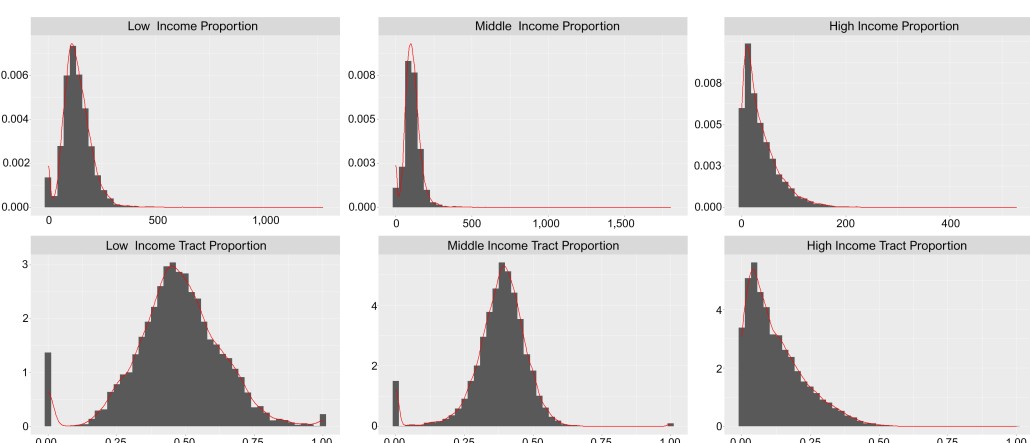

**Figure 2.** Distribution of income groups within the Australian Bureau of Statistics' Income Dataset.

### 4.1.2. Journey to Work

Commuting data was also obtained for the 2016 census year from the ABS. This refers particularly to the Journey to Work dataset, which includes the recorded number of individuals reporting to from a SA1-level 'Place of Usual Residence' to a Destination Zone (DZN) level 'Place of Work'. DZN areas are constructed from mesh block boundaries, but are not considered a statistical boundary [63]. The dataset comprises reported population flows in the entire Greater Sydney Metropolitan area. 11,171 SA1 and 2233 DZN entries were noted in the dataset. Approximately 1.21 million persons are enumerated within this data with respect to their individual flows, which equated to approximately 25 million individual trips between all origin and destination zones. A cursory inspection of the Journey to Work dataset reveals that the Sydney City Centre and its surrounding are the central locus of employment, with approximately 25 per cent of all flows. Other loci of employment with relatively high densities of population flow include Parramatta, North Sydney, Campbelltown, and Gosford. The distribution of flows show less variation across all other DZN areas.

The Journey to Work dataset was further extended to obtain both origin and destination capacities; and this can be derived from the sum of flows at both origin and destination zones. Their inputs within the developed accessibility model will be discussed in the subsequent section. The flow matrix was then restructured into a simple dataframe, which enabled the dataset to be joined by their respective zone codes to the appropriate SA1 and DZN boundary vectors. This data layer was segmented with proprietary vector data of house footprints obtained from Geoscape to identify the highest density cluster of houses within each delineated boundary. A median-weighted centroid was appended within each SA1 and DZN building cluster to obtain a single point for each origin–destination zone (Figure 3).

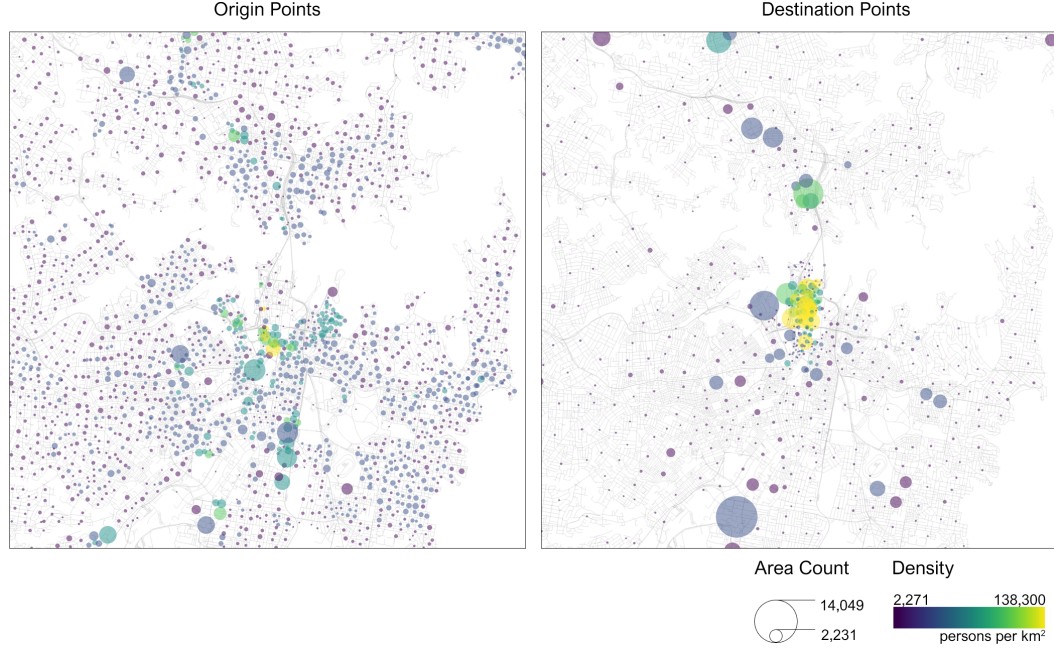

**Figure 3.** Example of origin and destination points centred on the CBD.

### 4.2. Road Network

The transportation network for the Greater Sydney region is from the GeoFabrik (Site Available at http://geofabrik.de/; accessed on: 28 June 2021) repository of OpenStreetMap (OSM) features on 5 August 2020. All OSM data layers within the Australian sub-region was selected, which included a separate vector dataset of all transport lines in the region. The dataset was then processed in two stages to obtain a functional road network for Greater Sydney. First, a spatial subset of the data was created with boundary files obtained

from the ABS. The vector boundaries were filtered by their individual region code attribute, '*1GSYD*', to return only data of the Greater Sydney region. The full OSM transport network was clipped to this boundary to obtain all line features within Greater Sydney (Figure 4). Next, the trimmed Greater Sydney transport network was then filtered through their attributes to obtain a network traversable by vehicles. These specific classes were defined from the metadata descriptions provided by Topf [64]. This process excluded line data for inappropriate transport modes such as bridleways, cycleways, and pedestrian only walkways. It should be noted that all unclassed transport links were also included in formation of the road network. The distribution of road type and the total road lengths can be found in Table 2.

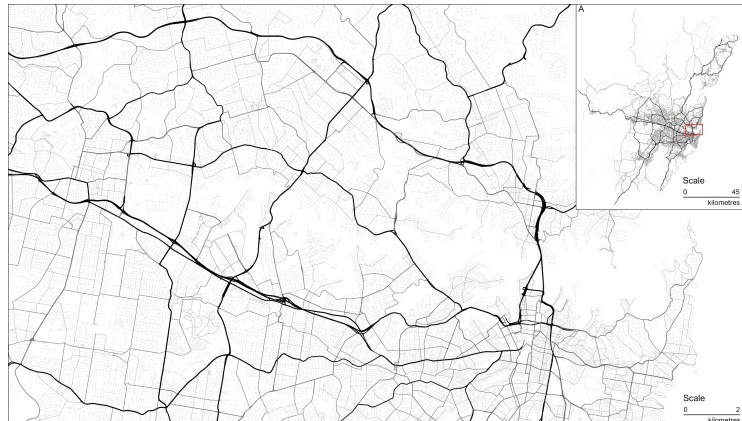

**Figure 4.** Cleaned road network.

**Table 2.** Length and number of roads, by classification, in Greater Sydney.

| Classification | Line Count | Total Length (km) |
|---|---|---|
| Motorway | 11,220 | 2466 |
| Primary | 10,305 | 1755 |
| Residential | 178,668 | 23,203 |
| Secondary | 16,705 | 2537 |
| Shared | 116 | 12 |
| Tertiary | 23,495 | 3141 |
| Tracks | 10,286 | 6530 |
| Unclassed | 8716 | 2363 |
| **Total** | **259,511** | **42,007** |

The obtained road network was processed to correct for any topological errors. This process involved the rectification of excessive vertices, overlaps, self-intersections, pseudo-nodes, and disconnected road islands. All line intersections were split to obtain all possible edge features given the study's lack of ancillary data to correct for all possible continuous throughways. This means some false intersections may have been introduced where two roads cross at different levels, however the number of such bridges is relatively small in Sydney, and all errors arising from the above specification would be equally imposed throughout the network. To ensure the full connectivity of the obtained road network, a reiterative intersection detection algorithm was developed and implemented to obtain all possible edge connections. In this algorithm, all 'motorway' features were used to reiteratively select all road intersections up to the boundaries of Greater Sydney. All unselected (i.e., disconnected road islands) clusters were discarded from the subsequent analysis. Figure 4 illustrates a small sub-region of the cleaned network.

The processed road network was used to calculate the network distance between each origin and destination point. A network-based distance calculation was preferred over Euclidean distances due to the anisotropic nature of the built environment. This is

particularly pertinent for Greater Sydney as the city contains many significant topological features that separate proximate areas. Figure 5 illustrates this problem, with the water bodies of Sailors Bay and Peach Tree Bay separating otherwise close suburbs.

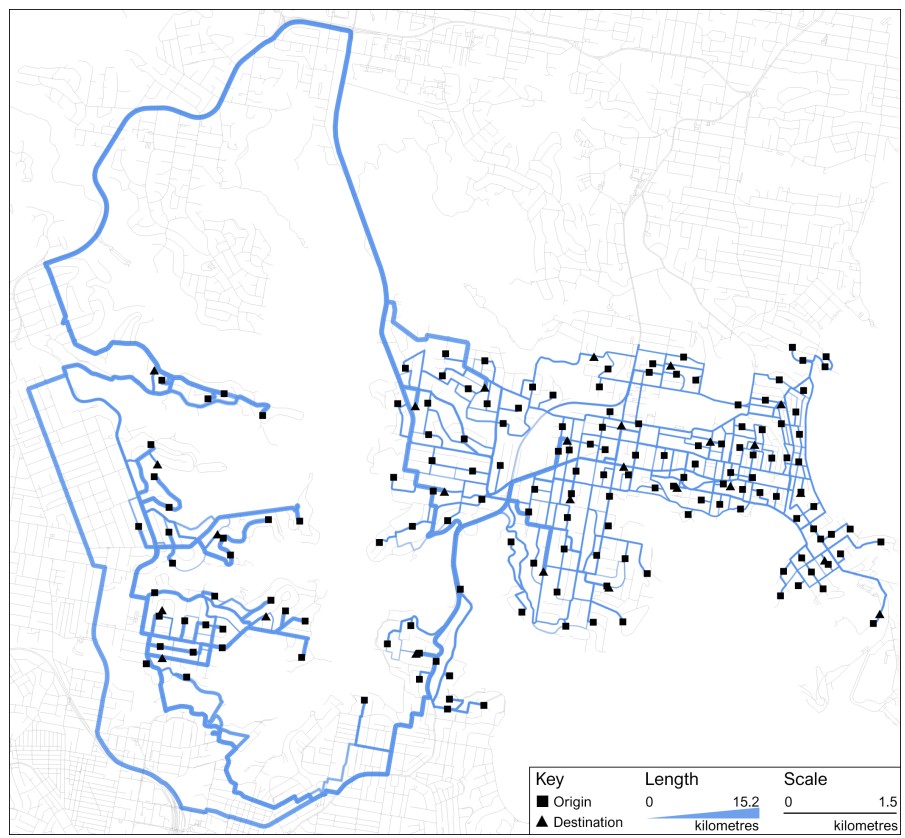

**Figure 5.** An example of the anisotropic nature of Sydney's road network.

### 4.3. House Price

The house price data used in this study was obtained from the Australian Property Monitor (APM). This database comprised of approximately 1.82 million property sales ranging between 2006 and 2020. Figure 6 illustrates the change of sale prices for houses over the above time range. A subset of the house prices transacted in the two quarters of 2016 was used, which included 32,068 house price data points. This date range was chosen to coincide with the census data used. The dataset includes a large number of individual variables for each property, with key information on the physical characteristics, temporal distribution of sales, and contract sale prices being particularly relevant to this study. These sales data included both house and unit (also known as apartment) sales—from which, only houses were used.

The dataset was processed in several rungs to negate issues of zero-inflation downstream; and to remove price outliers. Outliers were defined as those prices that are ≤2.5 per cent or ≥97.5 per cent within the house price range. All property entries were then geocoded using the open data Geocoded National Address File (G-NAF) available through Geoscape Australia.

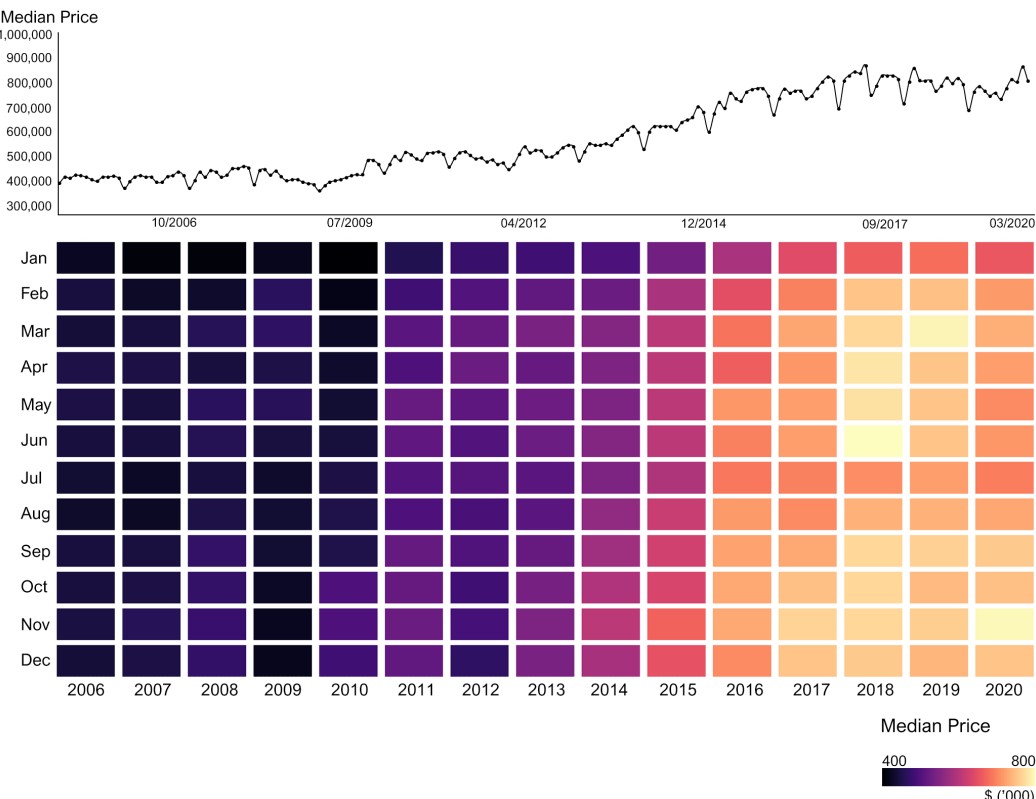

**Figure 6.** Temporal, count and price distribution of house prices from the Australian Property Monitor.

## 5. Methods

### 5.1. Entropy Measures

Entropy values are widely used as a statistical measure to describe specific attribute compositions within the urban system [65]. They are classically used to quantify disorder in an area; high entropy values equate to large variances within an area [65,66]. Neighbourhood entropy is a robust indicator of spatial variance; and it is widely applied in land-use analyses, population composition, and economic activity [66,67]. It is also used in segregation analyses to reflect the heterogeneity of populations in localised areas [68]. It offers a means to better understand diversity within a system. Here, the basic statistical form of entropy is adopted and applied to income groups.

The entropy measure utilised in this study can be represented by Equation (1),

$$H_i = - \sum_{j=1}^{k} P_{ij} \cdot ln(P_{ij})$$

(1)

where entropy (*H*) at location *i*, is determined by the proportion of a specific income group, *j*. *k* in the above equation refers to the number of income groups within the model. The proportion of this income group, $P_{ij}$, can be calculated with Equation (2),

$$P_{ij} = \frac{N_{ij}}{\sum_{j=1}^{k} N_i}$$

(2)

where the tract total of an income group, *j* in $N_{ij}$, is considered against the total of all income groups in the same tract, $N_i$. The maximum value of entropy ($H_{max}$) within a system is calculated with Equation (3)).

$$H_{max} = ln(k)$$

(3)

### 5.2. Accessibility Indices

This paper calculates accessibility using a gravity model [69], which can be expressed as Equation (4),

$$T_{ij} = k\frac{V_i^\mu W_j^\alpha}{f(d_{ij})} \tag{4}$$

where the interaction ($T_{ij}$) between zones $i$ and $j$ is determined by their respective 'mass' terms, $V_i$ and $W_j$. $k$ in the above equation refers the constant of proportionality as the interaction computed within the model is equal to the sum of their observed flows. $\alpha$ and $\mu$ are coefficients to be estimated. The 'mass' terms refer to known origin and destination specific attributes; and these may include variables such as total origin population or a factor for destination attractiveness (e.g., facility catchment [70], total floorspace [16], employment capacity [71,72], or quality of facility [73]). In this instance, the gravity potential model provides a more nuanced proxy of accessibility in Greater Sydney, given the available population and flow data from the ABS. The model's parameters and coefficients are computed using these datasets in the following way.

From the notation expressed in Equation (4), the gravity potential model can be alternatively written as Equation (5),

$$T_{ij} = kV_i^\mu W_j^\alpha d_{ij}^\beta \tag{5}$$

where the flows ($T_{ij}$) are derived in the same way from mass term and attraction factors noted at the origin ($V_i^\mu$) and destination ($W_j^\alpha$). It is worth reiterating that, the model is implemented using the JTW flows as $T_{ij}$; the total DZN density as $W_j^\alpha$; and the origin population total as $V_i^\mu$. Following Dennett's work [74], the natural log of each component within the model can thus be taken and respecified into in a log-linear format (ref. Equation (6)):

$$lnT_{ij} = k + \mu ln(V_i) + \alpha ln(W_j) - \beta ln(d_{ij}) \tag{6}$$

whereby, the model is considered as 'saturated' with all possible components to estimate potential flows. With this, the model can be fitted simultaneously to derive the necessary coefficients and factors for the relative attraction and emissivity of the origin and destination. However, considering the current specification, we also implemented a single production constraint to converge the model estimates accordingly to the origin totals ($O_i$) as seen in Equation (7),

$$\sum_j T_{ij} = O_i \tag{7}$$

Consequently, this allows the accessibility model to be reshaped into Equation (8),

$$\delta_{ij} = exp(\sigma_i + \alpha lnW_j - \beta lnd_{ij}) \tag{8}$$

where $\delta_{ij}$ refers to the observed flows; and $\sigma_i$ is taken as a categorical predictor for all flows in each origin SA1 (i.e., each individual SA1 code). Having now appropriately specified the model, the parameters for $\alpha$ and $\beta$ are solved for. The final parameter values used are 0.121 and $-2.278$, respectively. With obtained parameters, the production-constrained model can be calculated as denoted by Wilson [75] with the appropriate balancing factor, $A_i$. The preference for singly-constrained model over the attraction-production constraints is due to the unknown 'capacity' of employment within each DZN area; which, if constrained, would have been arbitrary. The calibration of the above parameters allows accessibility scores to computed with the classical gravity potential model (Equation (4)). The obtained values are then standardised between 1 and 0 for ease of interpretation.

The model performance is evaluated with the computed flow estimates against the observed flows. An acceptable $R^2$ of 0.61 was achieved, which suggests that the model is able to capture just over two-thirds of the employment flows occurring within Greater

Sydney. It is also worth noting that the above models utilise a power law as its distance-decay function ($f(d_{ij}) = d_{ij}^{\beta}$) over its exponential counterpart ($f(d_{ij}) = exp^{d_{ij}\beta}$). Both functions were previously tested against the JTW observed flows; hence, the choice of the power-law function given its better results.

### 5.3. Relationship Testing

Having now considered the above data and models, an Ordinary Least Squares (OLS) regression and a Geographically Weighted Regression (GWR) were conducted to better understand the components' intrinsic relationships within the urban system. The OLS method is a global fitting method, which computes a single parameter estimates for each independent variable. However, given the spatial nature of the house price, income, and accessibility data, these relationships should also be tested with respect to their possible spatial variations. The model is derived from the traditional linear regression models; however the GWR includes a bandwidth parameter that considers a kernel of neighbouring points for each observation. Generally, the GWR model can be expressed as Equation (9),

$$y_i = \beta_0(u_i, v_i) + \sum_j \beta_k(u_i, v_i)x_{ik} + \varepsilon_i \tag{9}$$

where the relationship between the dependent variable, $y_i$, and the set of independent variables, $j$, is represented by a continuous function $\beta_k(u_i, v_i)$ at different locations of ($u_i$, $v_i$) at observation $i$. Here, the location of each data point is represented by coordinates ($u_i, v_i$); $\varepsilon_i$ represents the residual variable of the model. The GWR requires a bandwidth parameter, $b$, to be set, which are either fixed or adaptive (i.e., variable values of $b$) kernels. A spatial weights matrix is then constructed, with points closer to location ($u_i, v_i$) assigned a higher value, and those exceeding the $b$ parameter are nullified. This can be generally expressed by Equation (10),

$$w_{ij} = \begin{cases} 1, & \text{if } d_{ij} < b \\ 0, & \text{otherwise} \end{cases} \tag{10}$$

where $w_{ij}$ denotes the individual weights for each location of ($u_i, v_i$); and $d_{ij}$ representing the distance threshold in consideration. It should be noted that the general expression for the above weights matrix is often altered through a weighting function, which is typically a Gaussian or bi-square function [76]. In this paper, an adaptive bi-square was opted for due to its more sensitive allocation of weights with the increasing distances of all other data points [76–78]. This function is denoted by Equation (11):

$$w_{ij} = \begin{cases} \left(1 - (\frac{d_{ij}}{b_{i(k)}})^2\right)^2, & \text{if } d_{ij} < b_{i(k)} \\ 0, & \text{otherwise} \end{cases} \tag{11}$$

In comparison to the general weights function where the $b$ represents a single integer, the $b_{i(k)}$ parameter here is variable with $k$ representing the number closest to point $i$ [79]. The optimum bandwidths are finally chosen through the minimisation of the Akaike Information Criterion (AIC) [76,79,80].

#### Multicollinearity Diagnosis

The above house price dataset was then linked to neighbourhood and point of interest datasets based on house location. A test for multicollinearity was conducted to reduce the redundancy and imprecision in the regression model coefficient estimates arising from these variables. The diagnosis was approached by calculating the variance inflation factor (VIF) between each variable with respect to house prices. VIF accounts for the change in the independent variable coefficient estimate against any correlations between all independent variables within the model [81]. A VIF that exceeds 10 is often considered problematic [81]. The tolerance factor considers the likelihood of each independent variable

being unaccounted for by all other independent variables within the model. Remedial measures to address the arising issues with multicollinearity include variable exclusions, principal component analyses, or step-wise regression applications [82]. This paper opted to exclude collinear variable pairs with an observed VIF scores greater than 10 (VIF $\geq$ 10).

## 6. Results Objective 1: Income Distributions in Greater Sydney

As the first objective, the spatial distribution of income groups within the Greater Sydney area was calculated. Entropy statistics ($H$) were used to quantify the spatial diversity between these income groups. These values ranged from 0.00 to a maximum of 1.10 ($H_{max}$)—meaning, in areas where $H = 1.10$, an equal distribution of all three income groups is observed. The global mean entropy value in Greater Sydney was calculated to be 0.862, with a minimum and maximum of 0.00 and 1.09, respectively.

Table 3 quantifies the correlation coefficients between all income groups and their respective entropy values. What is first noticeable is the negative relationship between the distribution of low-income groups in Sydney with their middle-($R = -0.22$) and high-($R = -0.56$) income counterparts. This relationship is also seen with respect to entropy values ($R = -0.22$). These findings come in stark contrast to those values associated with high- and middle-income earners, with both groups displaying similar correlation coefficients to the diversity index. These findings indicate that Sydney's income distribution appears to be highly asymmetric. Where it was initially thought that both high- and low-income groups would both display exiguous mixing (i.e., low entropy scores in both high- and low-income groups), this generalisation can only be extended to the latter. The results rather show that high-income groups in Sydney appear relatively more integrated at the metropolitan level, whereas low-income earners look to be concentrated in locations where those outside this income bracket remain unlikely to reside in.

**Table 3.** Global correlation coefficients between income groups in Greater Sydney.

|  | Low Income | Middle Income | High Income | Entropy |
|---|---|---|---|---|
| **Low Income** | 1.000 | −0.218 | −0.561 | −0.219 |
| **Middle Income** | −0.218 | 1.000 | 0.198 | 0.565 |
| **High Income** | −0.561 | 0.198 | 1.000 | 0.629 |
| **Entropy** | −0.219 | 0.565 | 0.629 | 1.000 |

This more cursory evaluation does hint towards a possible degree of socioeconomic segregation in Greater Sydney; however, a disaggregated view of income distributions is needed to investigate whether the prominence of low-income groups is a consistent mainstay of areas of low diversity. Figure 7 illustrates the spatial distribution of entropy and all income groups in view of this. It is evident from this visualisation that the spread of entropy values shows a relatively clear delineation between low- and high-diversity areas, with lower entropy areas concentrated in the Central and Western suburbs of Greater Sydney, such as suburbs around Fairfield, Merrylands, Auburn, and Mount Druitt. In these areas, the diversity index is estimated to range between 0.39 and 0.55; and they also have the highest proportions of low-income earners in Greater Sydney (Figure 8). Here, low-income groups constitute between 63 and 77 per cent of the population—often, with less than 2 per cent belonging to high-income groups (Figure 8). The findings echo back to previous work by Lee et al. [83], which discussed the divide of the Sydney's demographic and socioeconomic distribution along a 'latte line'.

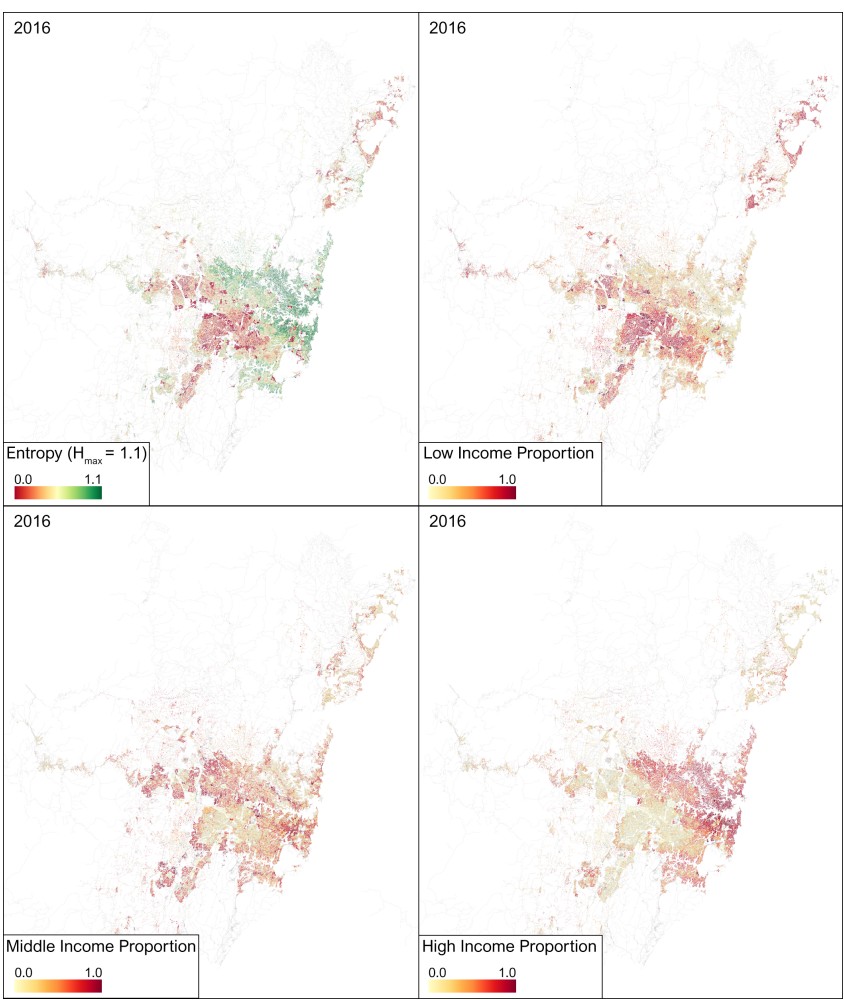

**Figure 7.** Entropy and income group distribution in 2016.

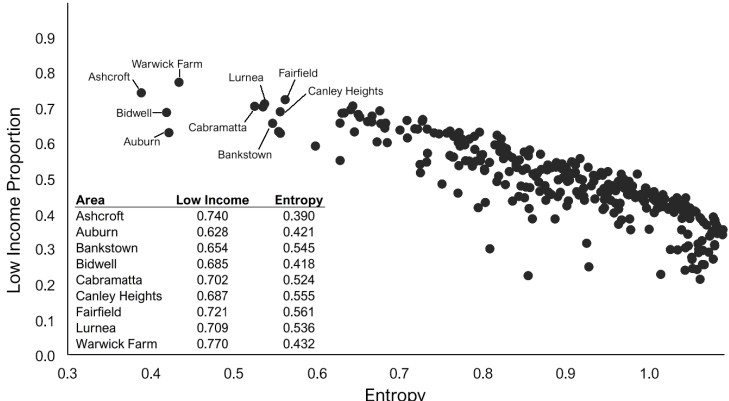

**Figure 8.** Entropy values against low-income groups in Greater Sydney.

When considered in line with the distribution of high-income groups, it is worth pondering on the reason these areas do not exhibit similar segregation seen with low-income areas. From the results displayed in Table 3, the distribution of high-income groups has already shown to be negatively correlated with the lowest earners ($R = -0.57$) in Sydney, which alludes to the spatial division between both groups. However, it is interesting to note that, despite this disparity, high-income groups also occupy those areas with the highest entropy values. One explanation is due to the asymmetrical size of the different groups—as the high income group represents only 12 per cent of Sydney's population overall, even areas where they are heavily over-represented will still show considerable

spatial overlap with the large group of middle-income earners, resulting in higher entropy values. For example, the highest *H* values are found in areas that tend to be more proximate to the Sydney Central Business District and coastal suburbs such as Mosman, North Sydney, the Eastern Suburbs, and Manly. In these neighbourhoods, approximately 33 per cent to 40 per cent of the population fall within the high-income bracket (Figure 9). Yet, they show a high entropy value, with the relative diversity being due to the spatial overlaps seen with middle-income earners ($R = 0.57$), who also tend to occupy these areas. Further work on affordable housing is required to better understand if barriers to residency still exist; particularly, if ongoing developments only cater for middle-income earners.

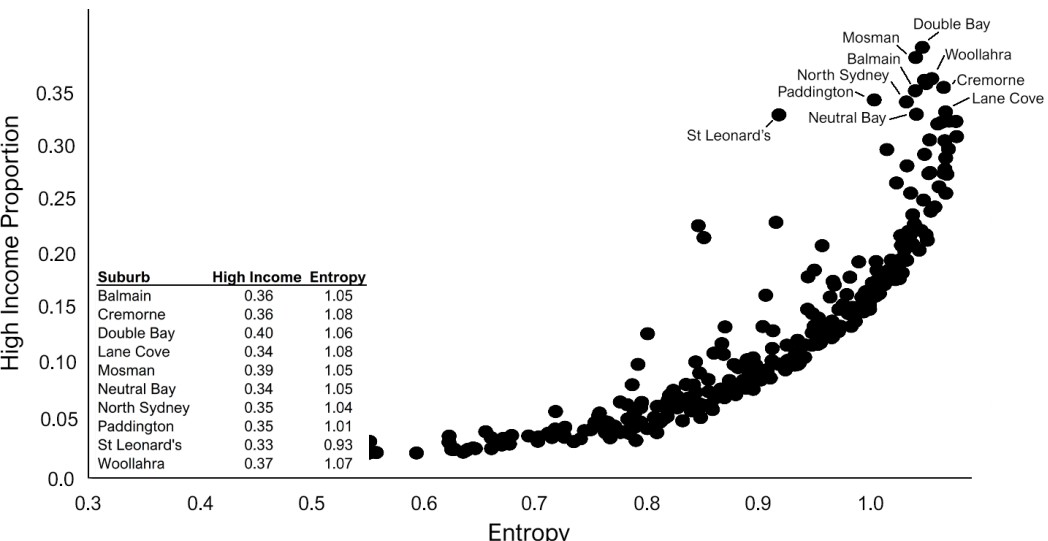

**Figure 9.** Entropy values against high-income groups in Greater Sydney

## 7. Results Objective 2: Accessibility and Income

The results above have quantified the spatial delineation between income groups in Sydney. They illustrate the possible sociospatial isolation that lower income groups may be confronted with given their distribution within Greater Sydney. They raise concerns as to whether these income groups may disproportionately experience the same negative spatial externalities seen in many cities globally. This paper thus seeks to investigate whether any variances in accessibility are observed in line with Greater Sydney's observed income spread. The distribution of accessibility scores, computed from the gravity potential model, are displayed in Figure 10. The normalised potential accessibility score ranged between 0.01 and 0.78, with a mean score of 0.07. The highest potential scores were noted in those suburbs within, and proximate to, Sydney's Inner City, North Sydney, Mascot, Macquarie Park, Epping, Chatswood, and Parramatta. These areas hold the highest employment densities concentrating approximately 40 per cent of all employment within the Greater Sydney area. Model estimates indicate that the Sydney Inner City area draws 22.5 per cent of all metropolitan employment movement. This accurately reflects the JTW census data, in which the area recorded a 22.7 per cent share of Sydney's total working population in 2016.

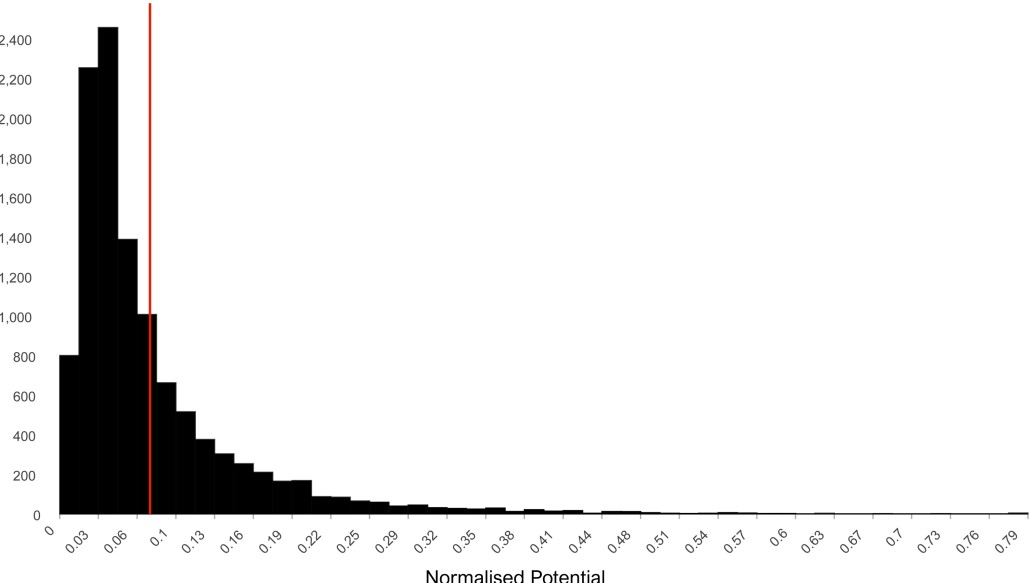

**Figure 10.** Distribution of computed potential accessibility scores to employments centres in Greater Sydney.

Conversely, the spatial distribution of lower potential accessibility scores appears relatively evenly throughout many suburbs across the Greater Sydney region. The median potential score of these areas is estimated to be approximately 0.04. This potential score comes despite smaller employment loci available (e.g., in Ashfield, Burwood, Balgowlah, Canterbury, Kogarah, Marrickville) within these largely suburban areas. The findings suggest that, despite these smaller pockets of employment, access to the Sydney City Centre, North Sydney, Macquarie Park, Parramatta, and Chatswood remains a far more significant determinant of employment accessibility within the Greater Sydney system. Several exceptions to this are noted within the Greater Sydney periphery however. In more distal areas, such as Katoomba, Campbelltown, and Gosford, potential accessibility scores remain relatively high. This deviation is likely attributed to large intervening distances between these areas to Sydney's primary employment hubs. As a result, these areas appear to have a secondary role in centralising employment flows within Sydney, particularly at the metropolitan's periphery. This centralising pattern of movement is also captured within the accessibility model's estimated flows. For example, between Sydney City and Campbelltown, there is a clear distinction between the source of movement within both these locations (Figure 11). Sydney City sees movement from all suburbs across Greater Sydney, whereas within Campbelltown, model estimates indicate predominantly inter-zonal movement, with more modest population flows from Camden, Liverpool, and Bankstown. In contrast, outer peripheral zones, such as the Central Coast, form a separate employment locus drawing from surrounding areas. In these areas, accessibility remains relatively high because employment flows are almost exclusively circumscribed to within the region (Figure 11).

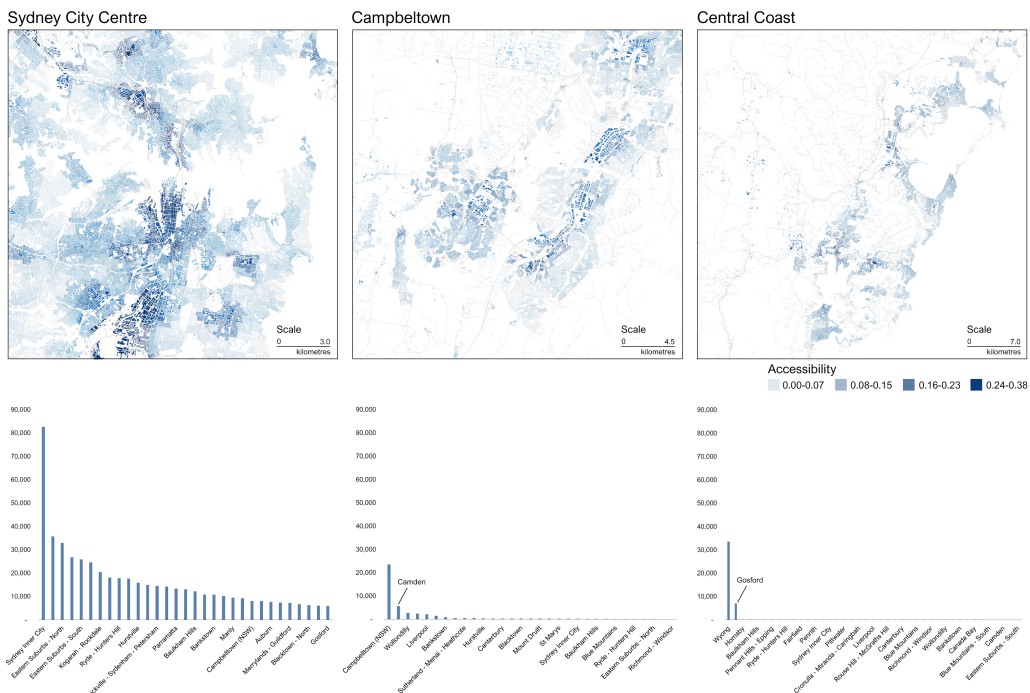

**Figure 11.** Sources of employment flows for major employment clusters.

Considering these flow trends in line with the spatial variances in accessibility, attention should be drawn to those intermediary zones that fall in between Sydney's major employment hubs, where low accessibility is noted throughout. These areas are typified by low density residential land-uses. The greater topological distances to Greater Sydney's employment centres may lead to disparate costs experienced by resident populations; thus, the need to better understand these accessibility costs in view of their socioeconomic structures. Figure 12 illustrates the distribution of accessibility scores relative to the distribution of low-income groups. It is interesting to note that, whilst low-income groups can generally be delineated within Sydney's inner South, Southwest, and Blacktown-area suburbs, there does not appear to be any clear correlation in accessibility variances within these areas ($R^2 = 0.01$). Therefore, the findings cannot support previous assumptions of socioeconomic stratification and isolation due to accessibility variance.

While employment accessibility appears to be experienced fairly equally across income groups within Sydney, it is worth noting that only employment density is taken as a proxy to employment area attractiveness. No measure of employment type or quality has been taken into account. It is also worth repeating that accessibility in Sydney has been shown to be pervasively low throughout the greater metropolitan area, with the exception of those areas adjacent to Sydney City, Chatswood, and Parramatta. Thus, whilst no spatial disadvantages can be attributed to lower-income areas in this study, further work is required to ascertain whether these disadvantages are seen perhaps in the quality and type of employment.

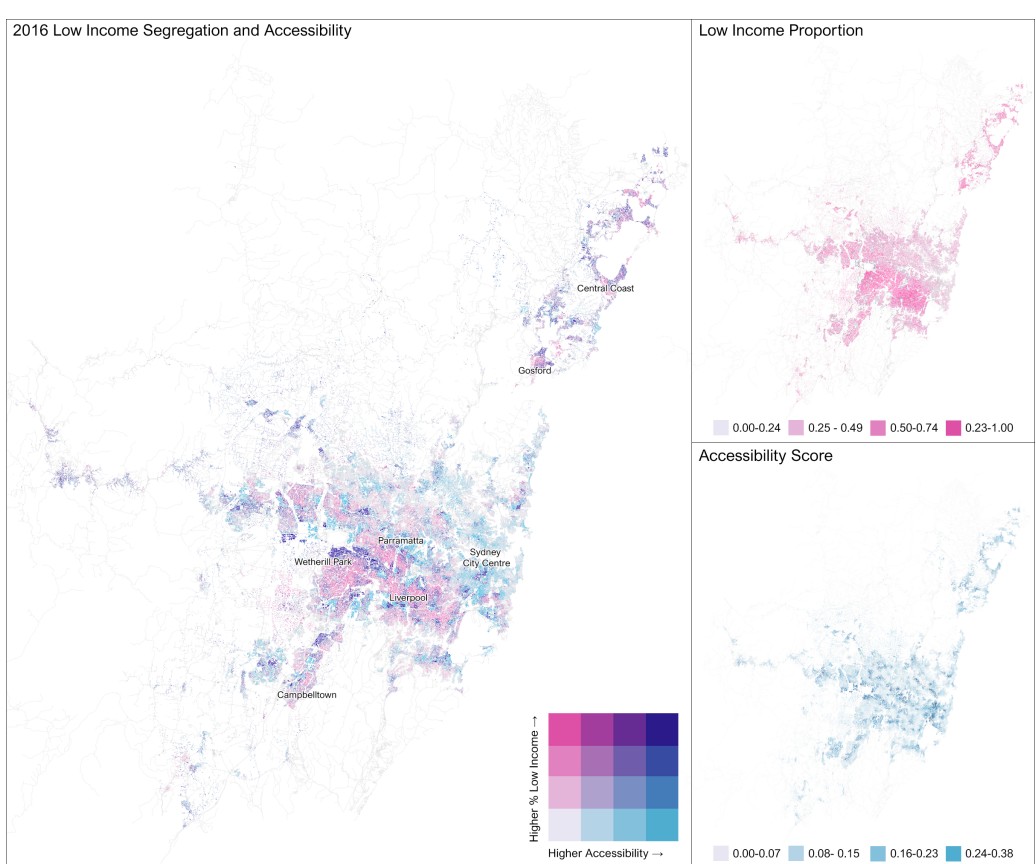

**Figure 12.** Bivariate Graphs: 2016 Accessibility and Income Segregation.

## 8. Results Objective 3: House Values, Accessibility, and Income Segregation in Sydney

### 8.1. Method

Having now considered the relationship between income groups and accessibility, this paper now shifts its focus towards understanding if these variables may also be reflected in house prices. First, an OLS regression was conducted to understand how house prices may be affected by these variables within the entire system. The results of the test are shown in Table 4. The model performance indicates a multiple- and adjusted-$R^2$ value of 0.81 for both, which indicates a fairly strong linear relationship across all variables whereby over 80 per cent of the model variances (normalised through a logarithmic transformation) can be accounted for by the independent variables.

### 8.2. OLS Results

For the most part, the results of the OLS model indicate a fairly predictable relationship between house prices and the independent variables used. House prices are noted to markedly increase with size. This was proxied by the number of bedrooms, bathrooms, and parking spaces, as well as increasing distances from major infrastructure features and major recreational sport facilities. This comes in addition to the negligible increases in house prices are also seen with increasing distances from educational facilities, and shopping centres. As also expected, negative changes in house prices are also linked to increased distance from Sydney City, beaches, and reduced crime rates.

It is interesting to note the differences in significance levels both income groups and accessibility scores display on house prices. The OLS model indicates that the presence of both low- and high-income groups are significant predictors for the variance in house prices in Greater Sydney, with the model indicating a significance level of at least 99 per cent. Accessibility scores are also significant at the global level, with a *p*-value of less than 0.01. Increasing proportions of low- and high-income groups display a relative change of −0.2 and 2.5 per cent change in house price, respectively. Further, at the metropolitan

level, there is a relative rise of 0.3 per cent of house prices in areas that have increased accessibility. It is worth noting that the presence of middle-income groups, as defined in this study, does not have the same level of predictive potential. This is likely due to their larger aggregation of income into a single group; and resulting relatively even distribution throughout all of Greater Sydney.

**Table 4.** Coefficient values derived from the OLS regression of variables to house prices.

| Residuals: | | | | | |
|---|---|---|---|---|---|
| | **Min** | **1Q** | **Median** | **3Q** | **Max** |
| | $-3.08558$ | $-0.14556$ | $-0.01844$ | $0.11633$ | $2.40827$ |
| **Coefficients:** | | | | | |
| | **Estimate** | **Std. Error** | **t-Value** | **Pr (>\|t\|)** | |
| (Intercept) | $1.33 \times 10^{1}$ | $5.69 \times 10^{-2}$ | 233.182 | $<2.00 \times 10^{-6}$ | *** |
| Bedrooms | $1.42 \times 10^{-1}$ | $1.58 \times 10^{-3}$ | 90.152 | $<2.00 \times 10^{-6}$ | *** |
| Bathrooms | $4.89 \times 10^{-2}$ | $1.48 \times 10^{-3}$ | 33.016 | $<2.00 \times 10^{-6}$ | *** |
| Distance to Sydney | $-1.45 \times 10^{-5}$ | $1.69 \times 10^{-7}$ | $-85.66$ | $<2.00 \times 10^{-6}$ | *** |
| Distance to Secondary City | $7.07 \times 10^{-7}$ | $1.74 \times 10^{-7}$ | 4.065 | $4.82 \times 10^{-5}$ | *** |
| Distance to Beach | $-2.44 \times 10^{-6}$ | $2.24 \times 10^{-7}$ | $-10.916$ | $<2.00 \times 10^{-6}$ | *** |
| Distance to Bus Interchanges | $2.91 \times 10^{-6}$ | $4.83 \times 10^{-7}$ | 6.031 | $1.65 \times 10^{-9}$ | *** |
| Distance to Highway | $7.50 \times 10^{-6}$ | $4.44 \times 10^{-7}$ | 16.883 | $<2.00 \times 10^{-6}$ | *** |
| Distance to Primary School | $1.02 \times 10^{-5}$ | $3.26 \times 10^{-6}$ | 3.118 | 0.00182 | ** |
| Distance to University | $1.12 \times 10^{-6}$ | $4.27 \times 10^{-7}$ | 2.617 | 0.00889 | ** |
| Distance to Swimming Pool | $-4.61 \times 10^{-6}$ | $9.32 \times 10^{-7}$ | $-4.952$ | $7.40 \times 10^{-7}$ | *** |
| Distance to Shopping Centre | $1.11 \times 10^{-5}$ | $7.50 \times 10^{-7}$ | 14.774 | $<2.00 \times 10^{-6}$ | *** |
| Distance to Sport Centre | $9.00 \times 10^{-6}$ | $7.27 \times 10^{-7}$ | 12.378 | $<2.00 \times 10^{-6}$ | *** |
| Population Above 65 (%) | $5.08 \times 10^{-3}$ | $2.17 \times 10^{-4}$ | 23.355 | $<2.00 \times 10^{-6}$ | *** |
| Crime Rate (%) | $-4.35 \times 10^{-2}$ | $1.62 \times 10^{-2}$ | $-2.681$ | 0.00735 | ** |
| Low Income (%) | $-1.87 \times 10^{-1}$ | $5.75 \times 10^{-2}$ | $-3.244$ | 0.00118 | ** |
| Middle Income (%) | $-1.02 \times 10^{-1}$ | $5.80 \times 10^{-2}$ | $-1.758$ | 0.0788 | . |
| High Income (%) | 2.51 | $5.97 \times 10^{-2}$ | 41.983 | $<2.00 \times 10^{-6}$ | *** |
| Accessibility Score | $2.73 \times 10^{-1}$ | $8.68 \times 10^{-2}$ | 3.142 | 0.00168 | ** |

Significance Codes: 0 '***' 0.001 '**' 0.01 '.' 0.1; Number of data points: 32,050; Residual standard error: 0.2455 on 32,049 degrees of freedom; Multiple R-squared: 0.8103, Adjusted R-squared: 0.8102; F-statistic: 7604 on 18 and 32,049 DF, *p*-value: $<2.2 \times 10^{-16}$; Residual sum of squares: 1931.751; AIC: 952.1797; AICc: 952.2059.

*8.3. GWR Results*

Whilst the OLS model provides a relatively sound representation of house price variances at the metropolitan level, local effects also need to be considered with respect to the spatial non-stationarity of property. These local effects are considered within the subsequent GWR model. The GWR model specified here shows an improvement in fit, with an adjusted $R^2$ value of 0.844 and a lowered residual sum of squares value of 1569. The spatial distribution of the local $R^2$ values, along with the coefficient estimates for accessibility, high- and low-income groups, is illustrated in Figure 13. The GWR model performance is high everywhere, with at least 67 per cent of house price variances accounted for across all of Greater Sydney.

A summary of the model's coefficient estimates is also provided in Table 5. The findings here provide a more granular estimate of the model's income and accessibility effects—in which, larger spatial disparities at local levels are more evident than previously indicated in the OLS model. The model's estimate coefficients indicate that, generally, improvements in accessibility garner a 3.3 per cent median increase in house prices. However, disaggregating these results spatially, the models show a clear divide between the East and West regions of Greater Sydney. The most striking observation from these GWR results is the difference in direction and magnitude of the effect of accessibility across Greater Sydney. Generally, there is a negative relationship of prices with accessibility in the far Southeast and Northeast of Sydney, whilst there is a weak positive relationship in central Sydney. Further, in Western and outer Sydney, a stronger positive relationship is documented. These findings suggest that accessibility may not necessarily be an important determinant of house prices in all areas. There may be several possible explanations for this.

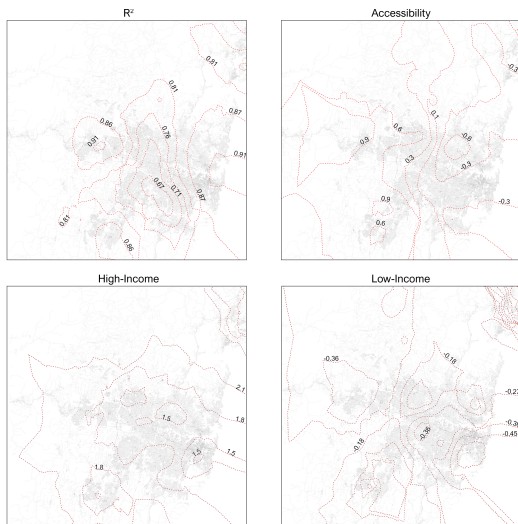

**Figure 13.** Contour visualisation of income and accessibility coefficients throughout Greater Sydney.

**Table 5.** Distribution of coefficient values from the GWR model of variables to house prices.

| | Summary of GWR Coefficient Estimates | | | | |
| --- | --- | --- | --- | --- | --- |
| | **Min** | **1Q** | **Median** | **3Q** | **Max** |
| Intercept | 6.2 | $1.3 \times 10^1$ | $1.4 \times 10^1$ | $1.4 \times 10^1$ | $1.8 \times 10^1$ |
| Bedrooms | $1.1 \times 10^{-1}$ | $1.2 \times 10^{-1}$ | $1.4 \times 10^{-1}$ | $1.6 \times 10^{-1}$ | $1.9 \times 10^{-1}$ |
| Bathrooms | $2.8 \times 10^{-2}$ | $4.0 \times 10^{-2}$ | $5.1 \times 10^{-2}$ | $5.8 \times 10^{-2}$ | $8.1 \times 10^{-2}$ |
| Distance to Sydney | $-3.4 \times 10^{-5}$ | $-3.0 \times 10^{-5}$ | $-2.3 \times 10^{-5}$ | $-1.5 \times 10^{-5}$ | 0.0 |
| Distance to Secondary City | $-1.6 \times 10^{-5}$ | $-4.7 \times 10^{-6}$ | $-5.6 \times 10^{-7}$ | $4.1 \times 10^{-6}$ | 0.0 |
| Distance to Beach | $-3.8 \times 10^{-5}$ | $-8.7 \times 10^{-6}$ | $2.3 \times 10^{-6}$ | $8.7 \times 10^{-6}$ | 0.0 |
| Distance to Bus Interchanges | $-3.8 \times 10^{-5}$ | $-9.7 \times 10^{-6}$ | $-3.2 \times 10^{-6}$ | $4.4 \times 10^{-6}$ | 0.0 |
| Distance to Highway | $-1.8 \times 10^{-5}$ | $3.0 \times 10^{-6}$ | $7.6 \times 10^{-6}$ | $1.1 \times 10^{-5}$ | 0.0 |
| Distance to Primary School | $-3.7 \times 10^{-5}$ | $1.3 \times 10^{-5}$ | $2.7 \times 10^{-5}$ | $4.5 \times 10^{-5}$ | $1.0 \times 10^{-4}$ |
| Distance to University | $-2.1 \times 10^{-5}$ | $2.3 \times 10^{-6}$ | $6.5 \times 10^{-6}$ | $1.1 \times 10^{-5}$ | 0.0 |
| Distance to Swimming Pool | $-2.7 \times 10^{-5}$ | $-1.3 \times 10^{-5}$ | -4.5 × $10^{-6}$ | $6.8 \times 10^{-6}$ | 0.0 |
| Distance to Shopping Centres | $-1.5 \times 10^{-5}$ | $-4.0 \times 10^{-7}$ | $7.2 \times 10^{-6}$ | $1.8 \times 10^{-5}$ | 0.0 |
| Distance to Sports Centres | $-1.2 \times 10^{-5}$ | $3.6 \times 10^{-6}$ | $6.7 \times 10^{-6}$ | $1.4 \times 10^{-5}$ | 0.0 |
| Population Above 65 | $1.2 \times 10^{-3}$ | $2.6 \times 10^{-3}$ | $3.7 \times 10^{-3}$ | $5.8 \times 10^{-3}$ | $8.9 \times 10^{-3}$ |
| Crime Rate | $-3.1 \times 10^{-1}$ | $-8.0 \times 10^{-2}$ | $-2.1 \times 10^{-2}$ | $1.0 \times 10^{-1}$ | $6.8 \times 10^{-1}$ |
| Low Income (%) | $-7.2$ | $-3.8 \times 10^{-1}$ | $-2.8 \times 10^{-1}$ | $-1.6 \times 10^{-1}$ | 5.0 |
| Middle Income (%) | $-6.6$ | $-3.1 \times 10^{-1}$ | $-1.8 \times 10^{-1}$ | $-6.7 \times 10^{-2}$ | 5.4 |
| High Income (%) | $-4.5$ | 1.5 | 1.7 | 1.8 | 7.7 |
| Accessibility Score | $-7.8 \times 10^{-1}$ | $-2.8 \times 10^{-1}$ | $3.3 \times 10^{-4}$ | $5.7 \times 10^{-1}$ | 1.1 |

Diagnostic information: Number of data points: 32,068; Effective number of parameters: 192.2415; Effective degrees of freedom: 31,875.76; AICc: $-5447.286$; AIC: $-5594.308$; Residual sum of squares: 1569.958; R-square value: 0.8458043; Adjusted R-square value: 0.8448744.

First, the higher attractiveness of homes in Sydney's more affluent suburbs (e.g., in Eastern suburbs) appear to be more sensitive to locational attributes such as views, access to green-spaces and beaches, good schools, and larger houses compared with employment accessibility. Importantly, these affluent suburbs are characterised by these better amenities compared with less prosperous suburbs [11]. In comparison, in more segregated areas in the outer Sydney suburbs, home choice becomes more utilitarian. In these areas, accessibility to employment becomes a more important factor in determining house prices. This is likely related to demographic differences seen between these areas. In wealthier areas of Sydney, accessibility based on journey to work may not be as important for wealthy retirees that cluster in these areas (e.g., Sutherland Shire, Hornsby-Warringah, and the Northern Beaches). Another issue may be the further anisotropic nature of Sydney's public transport network, resulting in travel times for subgroups of commuters that differ from the road network based accessibility used in this model. As shown in Figure 14, the Southeastern and Northeastern suburbs are relatively less served by rail lines or dedicated right-of-way bus services than other suburbs (in 2016—prior to the opening of new light rail in the Southeast).

This may mean that actual employment flows from the western suburbs are underestimated, and from the eastern suburbs are overestimated, in the accessibility model, which is based on road distances. This is a noted limitation of the current analysis with future work required to incorporate public transport accessibility into the model. Notwithstanding this, the developed accessibility model is relevant for a majority of Sydney's residents given the city's 65.2 per cent reliance on cars for commuting [84].

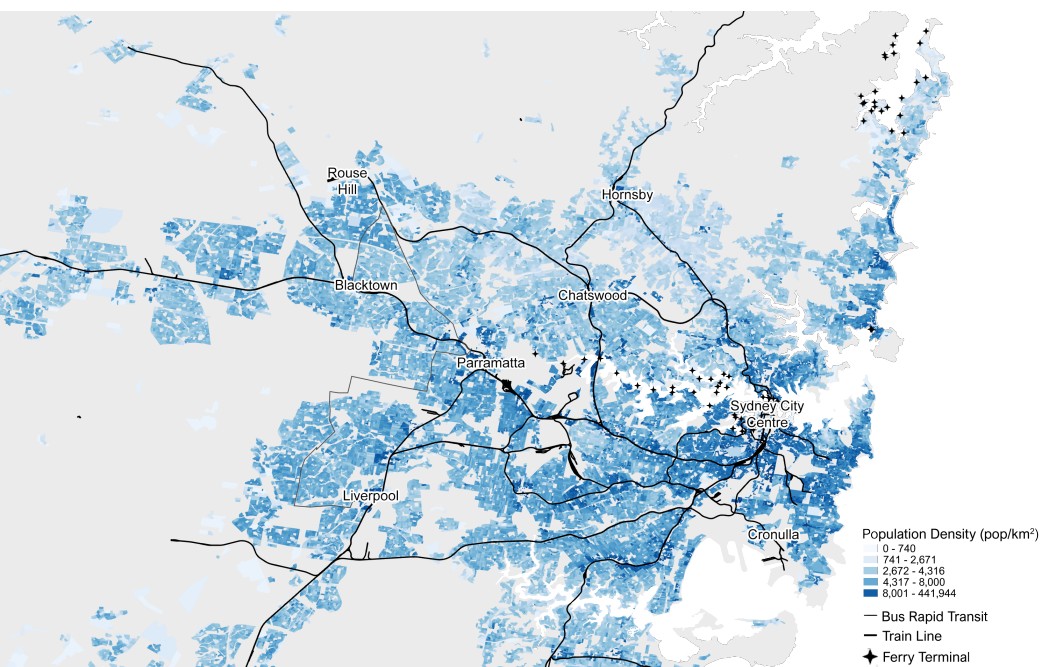

**Figure 14.** Major public transport lines and suburb population distribution in Sydney.

Rather, a stronger determinant of house prices in these areas may be the variable income group stratum present. The GWR results indicate a median decrease of 0.28 per cent associated within increasing proportions of low-income earners; whereas, a median increase of 1.7 per cent is noted with properties found in predominantly high-income areas. However, it is interesting to note the coefficient changes in areas such as Sydney's centre-east suburbs. In these relative higher priced areas, the GWR indicates that any small increase in low-income groups within the area has a much more detrimental effect on house prices than seen in the suburbs of Sydney's West and Inner West. The sensitivity of property values in these areas is indicative of the area's primacy. It suggests a level of income stratification in the area that is supported by raised house prices. Finally, whilst accessibility may not be the leading factor in determining property prices across Sydney, it is still considered a major factor in lower-income areas as represented by the model's coefficients (Figure 13). It can be interpreted that spatial equity remains an issue in Sydney, whereby the distribution of employment accessibility does not cover those groups that most need and want this accessibility.

## 9. Conclusions

This paper examined how and to what extent accessibility are related to income segregation and house prices in the Greater Sydney Area. Several key findings have been identified. First, low-income groups tend to be more clustered together in Sydney, which has given rise to enclaves of low-social diversity. However, this distribution to be asymmetric with higher income groups in the city. Middle- and high- income earners tend to be evenly distributed throughout the city, with increased demographic mixing. Second, poor accessibility does not appear to be a major contributor the income segregation phenomenon in Sydney. Accessibility remains relatively equal between both high-income and low-

income neighbourhoods given their proximity in Sydney. The results from this paper show that a relatively tenuous relationship between incomes and accessibility. In fact, in several areas, high income is accompanied by lowered accessibility. These findings suggest a level of self-isolation and lack of dependence on access to employment opportunities by higher-income residents. Lastly, it was also found that other individual property characteristics (e.g., the presence of views, access to better schools, and access to recreational amenities) in high income areas play a much larger role in deciding property values in these areas. This has been attributed to the dependency on personal vehicles and the lowered need for access to employment centres within the high-income stratum who may have alternative sources of income. This brings to light the main limitation of this study, where only road access is considered as a measure of distance. Whilst public transport is only used by approximately one third of Sydney's commuters overall, it is used by up to two-thirds in the high-income suburbs close to the city centre. Thus, employment access by public transport may be playing a role in the price of houses in these suburbs which is not accounted for here.

These findings have several profound implications for urban and housing policies. Specifically, policy makers should consider the issue of income segregation as low-income groups tend to be more segregated in their policy decision making. Further, enhancement of accessibility is not a feasible solution to address income segregation effectively as this is not a main contributor to the onset of income segregation in Sydney. However, future study can consider further work to incorporate multi-modal transport to provide a more holistic view of transport in Sydney. Further research could also consider accessibility not just to employment but accessibility to amenities and essential urban services.

These results were produced using data from before the COVID-19 pandemic, as 2021 census data was not yet available. It will be worth re-examining if these patterns change after COVID-19, noting that settlement and income patterns may take some time to settle into a stable form.

**Author Contributions:** Conceptualization, Matthew Kok Ming Ng; methodology, Matthew Kok Ming Ng; software, Matthew Kok Ming Ng and Josephine Roper; validation, Matthew Kok Ming Ng and Josephine Roper; formal analysis, Matthew Kok Ming Ng; investigation, Matthew Kok Ming Ng and Josephine Roper; data curation, Matthew Kok Ming Ng; writing—original draft preparation, Matthew Kok Ming Ng; writing—review and editing, Matthew Kok Ming Ng, Josephine Roper, Chyi Lin Lee, and Christopher Pettit; visualization, Matthew Kok Ming Ng and Josephine Roper; supervision, Matthew Kok Ming Ng; project administration, Matthew Kok Ming Ng. All authors have read and agreed to the published version of the manuscript.

**Funding:** This research received no external funding.

**Institutional Review Board Statement:** Not Applicable.

**Informed Consent Statement:** Not Applicable.

**Data Availability Statement:** Not applicable.

**Conflicts of Interest:** The authors declare no conflict of interest. The funders had no role in the design of the study; in the collection, analyses, or interpretation of data; in the writing of the manuscript; or in the decision to publish the results.

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
