# Peer review of "The Reflection of Income Segregation and Accessibility Cleavages in Sydney’s House Prices"

_ijgi, doi:10.3390/ijgi11070413_

Round 1

Reviewer 1 Report

This paper examines how accessibility and income segregation (as two main factors) influence the house prices in Sydney. Having reviewed the literature on the effects of accessibility and income segmentation on house price in sections 1 and 2, the data is introduced in details in section 4 and the methods in section 5. The results are presented and discussed in sections 6 to 8.

The paper is well-written and well-structured. The preliminaries are introduced and the literature is reviewed. The data and methods are well introduced, and finally the results are well presented and discussed. I am not an expert in the exact field and leave the scientific issues to other reviewers, but I recommend the paper for publication as it is.

My only recommendation is that authors move the detailed discussion of accessibility from the introduction to section 2.

Line 97: he ---> the

Author Response

Thank you so  much for your review and the time taken to read out manuscript. We have accepted all the changes proposed and below, we have provided the individual responses. 

Reviewer 1

Suggestion to move the detailed discussion of accessibility from section 1 to 2

Done as part of the restructure of section 1 & 2

Line 97 spelling error

fixed

Reviewer 2 Report

 This paper explores the relationship between accessibility, income segregation, and house prices in the greater Sydney area. This is a very interesting study. However, if the article wants to finally get published, it must be revised subversively. First, the current manuscript is more like a technical report than a paper. Secondly, the manuscript did not follow the standard format of the paper, nor did it use the template provided by the official. Third, the structure of the paper is confusing and illogical. So, I think the paper must be thoroughly revised if it wants to get final publication.

Author Response

Thank you so much for taking the time to review our manuscript. We really appreciate the effort and we have accept all changes. Specific responses to individual comments are listed below

Reviewer 4

The manuscript is more like a technical report than a paper

We are not entirely sure what the reviewer means by this as we have extensively discussed theory and background as well as the methods and results. We hope some of the following changes help to address the concerns.

the manuscript did not follow the standard format of the paper, nor did it use the template provided by the official

We have moved the paper into the journal’s template, including changing to numerical references. We have renamed the headings to clarify the format – there are multiple results sections owing to the extensive results obtained.

Third, the structure of the paper is confusing and illogical.

We have heavily restructured the first 2 sections and as noted renamed some sections to address this.

The abstract should not begin with “This paper explores ...”

Noted, we have rewritten the abstract

The logic is confused and the focus is not prominent. I don't know what message the author wants to convey to the reader in " Introduction "。Therefore, it must be completely rewritten.

2. The language in the article gives me a headache. The sentences and words in the article do not meet the standards of the paper and are too colloquial. This section must be rewritten if it is to be published

We accept the reviewer’s comments on the introduction. We have restructured it into a much shorter Introduction that lays out the problem area, with detailed accessibility material moved to the second section, now titled Literature Review.

We have tried to remove some of the wordy language that may have given the paper a colloquial appearance, for a tighter approach.

Line 22: What does "multifaceted" mean? And “built-environment”

Removed multi-faceted, have changed all built-environment to built environment, which we think is a common term in the urban field.

Lines 22-23: “It is a measure that is linked to numerous distinct, but, related, urban dimensions such as, land-use, network structures, and population distribution”    

I don't know what the author is trying to express?

“...but, related...” Is this expression appropriate?

Rephrased this sentence

Lines 22-38: The logic is confusing and I don't know what message the author is trying to convey to the reader. What is the meaning of this passage?

We have tried to clarify this section

Line 39:  “Perhaps more fundamentally...” Why "perhaps"? Is the author not sure? What do “its”,

“more systemic urban issues” mean?    

Line 40: Why does " understanding accessibility" provide an indispensable means?

                Is there no other way besides” understanding accessibility”? Why?

  Line 90:  “...of of...”?

.....

Noted, all have been changed

2 Accessibility, Income Segregation and House Prices:

Please completely change the structure

We have added headings to clarify that the purpose of this section is reviewing the literature on 3 subtopics (accessibility, segregation, and inequality in Australia specifically). We have tried to clarify the language and sentences throughout.

Research Objectives:

I suggest putting this section in "Introduction"

We have moved this section to the end of the (now shortened) Introduction

Data:

Please thoroughly check table 1-5

We have added a missing caption and removed some superfluous decimal places

Reviewer 3 Report

Dear Authors, 

The submitted article is very interesting and I like its structure.

What I miss is the step by step described methodology. 

Figures 8 and 11 are a bit unreadable. 

Did the authors notice the change in house prices caused by the COVID pandemic during their research? The research was conducted on data from 2016. Maybe it is worth considering comparing these results with, say, 2021.  

Author Response

Thank you so much for your review and the time taken to read our manuscript. We have accepted all changes, and have put the specific responses to your comments below.

Reviewer 2

Missing the step by step methodology

We have tried to tighten the text and use headings and subheadings to clarify where in the paper the methods and results for each objective are to be found

Figures 8 and 11 hard to read

We have submitted the highest possible resolution figures for the final version

COVID pandemic effects

We have included a suggestion for future work at the end (we wanted to use data that aligned temporally with our census data, and the post-COVID census data is not available yet)

Reviewer 4 Report

This is a very nicely written paper. It makes a nice contribution to the literature from both a methodological viewpoint but also in the examination of Sydney. Sydney's geographic nature provides additional points of interest due to the impact it has upon travel and commuting patterns. 

Author Response

Thank you so much for your kind review. We appreciate the time taken to read and summarise it. 

This manuscript is a resubmission of an earlier submission. The following is a list of the peer review reports and author responses from that submission.